# Health-Related Quality of Life According to Sociodemographic Characteristics in the South Korean Population

**DOI:** 10.3390/ijerph19095223

**Published:** 2022-04-25

**Authors:** Chan-Hee Park, Eunhee Park, Hyun-Min Oh, Su-Jin Lee, Sun-Hee Park, Tae-Du Jung

**Affiliations:** 1Department of Rehabilitation Medicine, Kyungpook National University Hospital, Daegu 41944, Korea; chany9090@gmail.com (C.-H.P.); ohm0105@gmail.com (H.-M.O.); 2Department of Rehabilitation Medicine, Kyungpook National University Chilgok Hospital, Daegu 41404, Korea; ehmdpark@knu.ac.kr (E.P.); sujin89898@gmail.com (S.-J.L.); 3Department of Rehabilitation Medicine, School of Medicine, Kyungpook National University, Daegu 41944, Korea; 4Division of Nephrology, Department of Internal Medicine, School of Medicine, Kyungpook National University, Daegu 41944, Korea; 00sum@hanmail.net

**Keywords:** health-related quality of life, RAND-36, South Korea

## Abstract

Health-related quality of life (HRQoL) concerns satisfaction with life and happiness with regard to physical, mental, and social factors. RAND-36 is a publicly available, self-administered questionnaire that examines eight health dimensions. This study evaluated the HRQoL of the South Korean population using the RAND-36 questionnaire and compared HRQoL across sociodemographic characteristics. From May 2015 to May 2019, South Koreans who visited public places aged 19–80 years were recruited and the RAND-36 questionnaire was administered. Overall, 1002 participants were recruited (mean age 45.34 years, 52% men). Men scored better than women in both physical and mental health (*p* < 0.05). There were significant differences in bodily pain (*p* < 0.05), general health perception (*p* < 0.05), and energy/fatigue (*p* < 0.05) dimensions according to the participants’ health condition. The HRQoL of South Koreans was lower than average in most dimensions compared with other countries. As the first study to assess this, its data can be used in future studies that apply RAND-36 to evaluate the HRQoL of diseased individuals, as they can compare their findings with those of our study population.

## 1. Introduction

The concept of health continues to change as medicine and the medical sciences develop. Health is defined as a dynamic state of human welfare characterized by physical, mental, and social potential that meets life-related needs as well as the absence of illness or infirmity. Health-related quality of life (HRQoL) relates to individuals’ life satisfaction or happiness with regard to their health status and social and cultural context [1]. This HRQoL mainly addresses public health issues in high-income and rapidly aging countries. HRQoL is typically measured using generic or disease-specific tools [2]. The decision to use a generic or disease-specific tool is determined by several factors, such as the reason for the measurement, the efficiency of the applicable tools, and the population in question. Generic measures are not specific to any disease, age, or treatment group and can be used to compare patients in different situations and with the general population [2,3].

General population data play an important role in determining whether groups or individuals’ scores are higher or lower than the average based on country of residence, age, and sex. Published general population data are available for several countries around the world, including the United States [4], Sweden [5], China [1,6], Hong Kong [7], Taiwan [8], and Japan [9]. However, to our knowledge, although HRQoL data of the general population are very important, there are no suitable general population data on HRQoL in South Korea. In particular, in a revolutionary dynamic situation such as the coronavirus pandemic period, there is a need for research on the HRQoL of the general population to compare between general populations and coronavirus survivors, or to confirm the change in HRQoL before and after coronavirus treatments. It is hoped that this study may be helpful for follow-up studies by presenting results prior to the coronavirus pandemic period, which will be the subject of comparison for future studies.

Many generic questionnaires have been developed to evaluate HRQoL; one of the most widely used instruments is the Short Form-36 Health Survey (SF-36) [1]. The SF-36 was created in the Medical Outcome Study (MOS), a four-year study that specifically focused on outcomes of care [10]. Since its creation, many variations and derivatives of the SF-36 have been developed. In particular, working with the RAND Corporation in 1992, Ware and Sherbourne published their version of SF-36, known as Ware-36, which focuses on eight health domains [11,12]. In 1993, Hays, Sherbourne, and Mazel released the RAND-36 [13]. Unlike Ware-36, the RAND-36 questionnaire and scoring methods are publicly available on the RAND Corporation’s website. A longitudinal study performed by MOS showed that the difference in scores between the Ware-36 and RAND-36 scales was subtle, without any significant differences [13]. Specifically, the two surveys differed only in terms of bodily pain and general health dimension subscales. However, according to Hays et al., bodily pain and general health dimension scores had a correlation coefficient of 0.99 [13]. 

Considering this, we aimed to obtain HRQoL-related data for South Korea using the RAND-36 survey and identify the sociodemographic factors that influence HRQoL.

## 2. Material and Methods

### 2.1. Data Collection

This retrospective cross-sectional study was conducted in South Korea to evaluate HRQoL in five urban and suburban areas. A total of 1002 South Korean citizens aged 19–80 years were recruited to visit public places where areas had foot traffic, including community centers and train stations that attract large populations. This study used a face-to-face questionnaire in Korean. Informed consent was obtained from all of the participants involved in this study. The inclusion criteria were those 19 years of age or older who are capable of independent daily living. From May 2015 to May 2019, a total of 1002 South Koreans were recruited. The participants were asked to fill out a questionnaire that included the RAND-36 health survey and items designed to obtain their general information, such as age, sex, region of residence, education status, occupational status, and marital status. In addition, respondents were asked whether they had been diagnosed with the following diseases: hypertension, angina, myocardial infarction (MI), diabetes mellitus (DM), or cancer and whether they had any of the following health problems: allergies, back pain, visual impairment, skin problems, chronic lung problems, hearing problems, functional impairment in a leg or arm, or joint pain. 

Responses to items concerning diseases and current health problems were classified into four categories: (1) no diseases and current health problems, (2) past or current diseases only, (3) current health problems only, and (4) diseases and current health problems [14]. The respondents took approximately 10–15 min to complete the questionnaire, immediately checked the questionnaire after completion, and if data were missing, they were asked to enter the data in question. If there were missing data even after this, the questionnaires were excluded from the statistical analysis. Ethical approval was obtained from the Institutional Review Board of Kyungpook National University Chilgok Hospital (KNUCH IRB no. 2018-11-020).

### 2.2. RAND-36

The RAND-36 is a standardized questionnaire, perhaps the most widely used health-related quality of life (HRQoL) survey tool in the world today [15]. It includes eight health dimensions: physical function (PF), role limitations due to physical health problems (RP), bodily pain (BP), general health perception (GHP), energy/fatigue (EF), social function (SF), role limitations due to emotional problems (RE), and emotional well-being (EWB). According to the instructions provided on the RAND Corporation website, the eight health dimensions were calculated using 36 items (https://www.rand.org/health-care/surveys_tools/mos/36-item-short-form.html, accessed on 5 May 2015).

The items evaluating role limitations (physical and emotional) are answered using “yes” or “no” responses, whereas the other items are scored using scales ranging from 3 to 6 points. According to RAND-36 scoring algorithms, the raw scores for each item should be summed and converted to a 0–100 scale, with 100 representing the best possible health state [13]. The RAND-36 dimensions can be classified into two categories: physical health composites (PHCs), which relate to physical function, and mental health composites (MHCs), which relate to mental and emotional well-being. PHCs were obtained by summing the scores for the PF, RP, BP, and GHP dimensions, whereas MHC was determined by summing the scores for the EF, SF, RE, and EWB dimensions [13].

### 2.3. Statistical Analysis

Descriptive statistics are presented as mean values and standard deviations (SDs) and included percentages for absolute counts regarding age, sex, education level, marital status, and health condition. The internal consistency of the RAND-36 questionnaire was evaluated using Cronbach’s α coefficient; a Cronbach’s α of 0.7 or greater is generally considered to indicate adequate internal consistency [16]. In addition, construct validity was evaluated using correlation analysis, and factor loadings greater than 0.50 were considered to indicate validity [17]. Finally, analysis of covariance was used to assess the independent effects of education, marital status, and health condition on the RAND-36 scale by controlling for age and sex effects. The data were processed and analyzed using a statistical software program (IBM SPSS Statistics 25.0, IBM, New York, NY, USA).

## 3. Results

### 3.1. Sample Characteristic

A questionnaire survey was administered between May 2015 and May 2019. Data from 1002 participants (525 men and 477 women) were analyzed. All subjects were aged between 19 and 80 years, with an average age of 45.34 ± 12.00 years. Respondents’ characteristics are listed in Table 1.

Of the 1002 subjects, 26% had a past or current disease, with hypertension being the most common (160; 63%), followed by MI (16; 6%), angina (10; 4%), DM (65; 26%), and cancer (61; 24%). Approximately 30% had a current health problem, with chronic back pain (115; 39%) and chronic allergies (89; 30%) being the most common.

### 3.2. Data Quality

The internal reliability of the RAND-36 was assessed using Cronbach’s α coefficient. This indicated that six of the eight dimensions had good internal consistency, with the exception of EF and EWB. Overall, of the eight dimensions, the lowest Cronbach’s α coefficient was found for the EWB (0.366), whereas the SF scored the highest (0.989; Table 2).

Spearman’s correlation analysis was used to analyze the correlation between the dimensions and items. PF, RP, BP, and GHP were more closely related to the PHC scale, whereas EF, SF, RE, and EWB were more closely related to MHC. Moreover, the correlation between PHC and its constituent dimensions ranged from 0.577 (PF) to 0.710 (GHP), whereas the correlation between MHC and its component dimensions ranged from 0.580 (RE) to 0.875 (EWB; Table 2).

### 3.3. HRQoL Data

The HRQoL data stratified by age and sex are presented in Table 3. For each dimension, when all age groups’ scores were considered cumulatively, men scored higher than women (*p* < 0.05). In this regard, the difference in the RE dimension was greatest at 4.35, and the difference in the EWB dimension was smallest at 1.96 points on a scale ranging from 0 to 100. Comparing specific age groups across sexes, females had lower scores than males in almost all dimensions. Here, the exceptions were PF, SF, and RE for females aged 30–39 years, GHP and SF for females aged 19–29 years, and EWB for females aged 50–59 years.

Among males, subjects aged 50–59 years showed the lowest scores in PF, GHP, and EWB, and those aged 40–49 years showed the lowest scores in RP, BP, and RE. In EF and SF, subjects aged 30–39 years had the lowest scores, whereas, in the PHC scales, subjects aged 60 years or older had the highest scores, and subjects aged 50–59 years had the lowest scores.

Among females, subjects over 60 years old returned the lowest scores in PF, RP, BP, GHP, and EF, whereas subjects aged 20–29 years showed the lowest scores in the EWB dimension. For the MHC scale, subjects aged 30–39 years returned the highest scores, whereas subjects aged 40–49 years showed the lowest scores. For the PHC scales, the lowest scores were provided by participants over 60 years of age; the higher the age, the lower the score. 

When age and sex were adjusted for, marital status was found to have no significant effect on any dimension (Table 4). However, when comparing raw scores, widow/widower subjects (15; 1%) showed the highest scores in all dimensions except for PF, in which divorced/separated subjects (11; 1%) showed the highest score. However, divorced/separated participants had the lowest scores in all dimensions except PF, RP, and EWB. Education level had no significant effect on any of the dimensions. In all dimensions, subjects who only had elementary school education (45; 4%) showed the lowest scores; however, there was no corresponding pattern among subjects with high education levels (i.e., they did not return the highest scores in all dimensions).

For BP, GHP, and EF, after controlling for the effects of age and sex, we found statistical differences in health conditions (Table 4). As expected, subjects without diseases and current health problems had the highest scores, whereas respondents with no diseases but current health problems returned the lowest scores. Health conditions had no significant effect on the MHC scale but had a significant effect on the PHC scale, which seems to reflect the results for BP and GHP.

## 4. Discussion

This study observed HRQoL-related data for South Korea using the RAND-36 health questionnaire. Such data can provide a basis for future studies seeking to analyze and evaluate the quality of life of South Koreans with regard to age and sex. Additionally, the HRQoL data can be used as comparative data for future studies that seek to use RAND-36 to evaluate the HRQoL of patients with specific diseases.

To perform this, internal consistency was evaluated using Cronbach’s α coefficient, and we consequently determined that six of RAND-36’s eight dimensions had good internal consistency, with EF and EWB being the exceptions. Specifically, the Cronbach’s α coefficients for the EWB and EF were 0.366 and 0.369, respectively. This low coefficient can be attributed to the characteristics of South Koreans. Traditionally, South Koreans are reluctant to reveal the state of their mental health [18,19]. For example, among South Koreans, many mental health patients choose to keep their treatment and condition secret. Socially, the country has a culture that regards the diagnoses of mental illnesses as different from those of physical illnesses. For items 24, 25, and 28, which are part of the EWB dimension and concern the negative aspects of mental health, the respondents may have selected responses that portrayed their mental health more positively, regardless of their actual condition. In contrast, for items with positive content, such as items 26 and 30, the respondents may have selected responses reflecting their reality. For EF, participants may have given more positive responses to items 29 and 31 and may have answered positive items, such as 23 and 27, more accurately. This should be understood as a characteristic of regions with significant differences in the diagnosis rate of depression, including East Asia, or regions, such as South Korea, with low access to mental health services for various reasons (social awareness, institutional disadvantage, treatment costs, drug side effects). These characteristics should be considered when interpreting the study results.

Our results showed that the RAND-36 questionnaire has reliable construct validity, which is consistent with the findings of previous surveys [20]. Our correlation analysis indicated that each of the eight dimensions was highly correlated with the related composite scale. Specifically, we found that PF, RP, BP, and GHP had higher correlations with PHC, whereas EF, SF, RE, and EWB had higher correlations with MHC, which is consistent with previous studies [11,13,20]. Thus, using RAND-36 to evaluate the HRQoL of the general South Korean population was determined to be reasonable. 

Previous studies have shown that men generally scored higher than women in terms of quality of life index by age group. In particular, the higher the age group, the larger the gap, which shows that there are time and economic reasons behind the decline in quality of life as people age [21,22]. It turns out that time factors are at play in middle age and economic factors are at play in old age.

In this study, quality of life was divided into PHC and MHC groups. For the PHC dimensions, male respondents scored higher than female respondents. For females, physical scores tended to decrease with age, which can be attributed to age-related biological changes, and there was no change in their physical role or position despite being older. In males, physical scores decreased up to the 50–59 years age group and then increased, with better health status observed among older individuals. This may be due to sociological changes that occur in men when they retire in their early 60s. At that time, they can pay more attention to their health because of the economic and time allowance. 

Regarding MHC dimensions, as with the PHC scales, males scored higher than females. Men had the poorest mental health at 30–39 years, when they were likely to be in the most stressful situations. At this age, they must find jobs and struggle to survive in employment. They also establish new homes and support themselves, both economically and mentally. The finding that older individuals had the best mental health was attributed to the above-mentioned sociological changes that occur at retirement. Sprangers and Schwartz [23] suggested that men over 60 years of age scored higher on the PHC and MHC scales as a result of the theoretical model of response shift. Unlike males, females aged 40–49 years had the lowest scores on the mental scale. At this age, middle-aged women may have a very poor quality of life, not only because of family problems but also because of children’s educational problems. 

Education level and marital status had no significant effects on any dimension. In South Korea, education up to middle school is compulsory, but South Koreans are generally zealous about educating their children, with a high school enrollment rate of 99% and university enrollment rate of 69.76%, which is the highest among OECD countries [24,25]. Among the participants in this survey, 91% had graduated from high school. The fact that marital status did not have a significant effect was attributed to the Confucian culture that persists in South Korea. In South Korea, divorce and separation are considered sinful and worthy of social punishments. In addition, there are many cases where couples refrain from divorce or separation to avoid negatively affecting their children. Reflecting on these Confucian characteristics, 97% of the respondents in this study were single or married/cohabitant. These social and cultural characteristics significantly affected education level and marital status. 

This study had some limitations. First of all, we did not conduct sampling before recruiting our study population. Random sampling is the best method for evaluating the quality of life in participants. However, this was not done in our study, and there is a possibility of sampling bias. Second, we recruited participants and interviewed them face-to-face in the foot traffic areas that attract large populations, so it took a long time, namely four years. This could cause a source of bias. Third, among the total respondents, the number of respondents in the widow/widower and divorced/separated groups in terms of marital status was very small, and the number of respondents in the elementary and middle school education groups in terms of education level was very small. This was due to socioeconomic and cultural differences, which may have affected the statistical analysis. To recruit respondents from various areas, questionnaires were distributed to public and commercial service organizations, community centers, and train stations that attract large populations. Among the respondents recruited, some lived in suburban areas; however, as most of the areas were located downtown, we may have recruited more urban-based than suburban-based people, which may have affected the results. Although the interviewers who instructed the participants to complete the questionnaire received identical instructions, there could have been differences in the explanations provided by the interviewers, which might have influenced the results. It is difficult to evaluate this quantitatively; however, it can still be considered a limitation of this study.

## 5. Conclusions

In conclusion, the HRQoL data for the South Korean population showed that the scores were somewhat lower than those of other countries or cities. In particular, South Korea’s EF and EWB dimensions, which are part of the MHC, were the sixth lowest among the countries and cities analyzed. There were no significant differences in HRQoL scores with regard to education level and marital status in South Korea, but there were significant differences in BP, GHP, and EF with regard to health conditions. These HRQoL data for our study population can be used in future studies that seek to use the RAND-36 questionnaire to evaluate the HRQoL of patients with specific diseases and compare the results with those of our study population. In addition, this is a kind of preliminary study in a study targeting a general population that has been subjected to appropriate random sampling in the future, and it is expected that it will be helpful in interpreting the results.

## Figures and Tables

**Table 1 ijerph-19-05223-t001:** The demographic characteristics of the respondents (n = 1002).

Sociodemographic Factors		n (%)
Age	29 or less (years)	122 (12%)
	30–39	197 (20%)
	40–49	302 (30%)
	50–59	264 (26%)
	60 or more	117 (12%)
Gender	Male	525(52%)
	Female	477(48%)
Education level	Elementary school education	45 (4%)
	Middle school education	47 (5%)
	High school education	330 (33%)
	University education	580 (58%)
Marital status	Single	184 (18%)
	Married/Cohabitant	792 (79%)
	Widows/Widowers	15 (1%)
	Divorced/Separated	11 (1%)
Health condition	No disease and current health problem	525 (52%)
	Past or current disease only	180 (19%)
	Current health problem only	224 (22%)
	Disease and current health problem	73 (7%)

**Table 2 ijerph-19-05223-t002:** Reliability estimates (Cronbach’s α) and correlation for the RAND-36 dimensions.

		Reliability	Correlation		
Dimension	No. of Items	Cronbach’s α	Correlations between Dimensions and Items	PHC	MHC
PF	3, 4, 5, 6, 7, 8, 9, 10, 11, 12	0.875	0.299–0.756	0.577	0.380
RP	13, 14, 15, 16	0.904	0.812–0.869	0.707	0.452
BP	21, 22	0.774	0.820–0.945	0.648	0.397
GHP	1, 33, 34, 35, 36	0.785	0.612–0.808	0.710	0.570
EF	23, 27, 29, 31	0.369	0.618–0.711	0.488	0.757
SF	20, 32	0.989	0.761–0.909	0.538	0.737
RE	17, 18, 19	0.878	0.865–0.882	0.548	0.580
EWB	24, 25, 26, 28, 30	0.366	0.604–0.740	0.459	0.875

PF, physical functioning; RP, role limitation, physical; BP, bodily pain; GHP, general health perception; EF, energy/fatigue; SF, social function; RE, role limitation, emotional; EWB, emotional well-being; PHC, physical health composites; MHC, mental health composites.

**Table 3 ijerph-19-05223-t003:** Mean RAND-36 dimensions scores (SD) by gender and age groups (higher scores indicate better health).

	Age Group											
	≤29 Years		30–39 Years		40–49 Years		50–59 Years		≥60 Years		ALL	
	M (n = 49)	F (n = 73)	M (n = 100)	F (n = 97)	M (n = 152)	F (n = 150)	M (n = 152)	F (n = 112)	M (n = 72)	F (n = 45)	M (n = 525)	F (n = 477)
PF	92.55	92.12	90.10	90.67	91.12	88.83	88.62	85.80	91.88	78.22	90.44	88.00
	(10.01)	(12.27)	(15.06)	(11.03)	(14.84)	(12.56)	(18.22)	(16.50)	(13.01)	(21.32)	(15.36)	(14.72)
RP	86.73	79.79	84.50	83.76	75.33	74.67	79.61	75.00	82.64	62.78	80.38	76.26
	(33.50)	(36.48)	(32.33)	(30.63)	(40.07)	(35.12)	(35.98)	(35.36)	(36.25)	(44.79)	(36.46)	(35.85)
BP	84.63	84.05	84.04	79.39	82.69	78.93	82.74	80.21	86.58	73.27	83.68	79.58
	(16.95)	(18.06)	(19.06)	(19.08)	(19.65)	(21.73)	(20.64)	(21.37)	(16.53)	(24.25)	(19.18)	(20.94)
GHP	62.00	62.56	62.19	61.40	62.84	59.03	61.43	60.21	65.97	57.27	62.66	60.16
	(16.02)	(20.40)	(17.99)	(18.32)	(15.60)	(17.32)	(15.91)	(16.62)	(17.48)	(20.82)	(16.47)	(18.20)
EF	57.96	53.49	56.25	54.28	58.16	54.53	57.14	54.78	59.58	53.33	57.68	54.27
	(12.45)	(15.76)	(13.36)	(14.98)	(13.41)	(13.56)	(13.32)	(14.48)	(13.76)	(17.58)	(13.32)	(14.77)
SF	84.95	86.13	84.13	86.08	85.53	78.75	85.12	83.48	89.24	80.00	85.60	82.60
	(16.33)	(20.15)	(20.09)	(18.56)	(18.32)	(19.90)	(19.34)	(19.29)	(15.60)	(19.84)	(18.45)	(19.70)
RE	85.03	78.54	83.33	84.19	77.85	71.78	82.89	78.27	82.87	77.04	81.71	77.36
	(31.23)	(37.42)	(34.33)	(30.84)	(38.92)	(40.27)	(33.88)	(35.71)	(38.35)	(41.33)	(35.85)	(37.23)
EWB	62.69	60.33	63.20	62.47	64.08	61.01	62.66	62.71	66.17	62.04	63.66	61.70
	(11.06)	(15.34)	(13.59)	(13.95)	(13.91)	(13.54)	(13.26)	(14.61)	(14.30)	(14.11)	(13.47)	(14.19)
PHC	52.55	51.83	52.01	51.34	51.15	49.86	51.10	49.86	52.84	46.57	51.66	50.15
	(5.10)	(7.19)	(6.68)	(6.36)	(6.98)	(6.97)	(7.10)	(7.48)	(6.00)	(9.50)	(6.69)	(7.39)
MHC	47.35	45.93	47.02	46.85	47.43	44.90	47.13	46.33	48.91	45.42	47.46	45.84
	(6.13)	(8.46)	(7.66)	(7.33)	(7.22)	(7.70)	(7.42)	(7.73)	(6.90)	(8.10)	(7.23)	(7.80)

PF, physical functioning; RP, role limitation, physical; BP, bodily pain; GHP, general health perception; EF, energy/fatigue; SF, social function; RE, role limitation, emotional; EWB, emotional well-being; PHC, physical health composites; MHC, mental health composites.

**Table 4 ijerph-19-05223-t004:** RAND-36 dimensions scores in relation to marital status, education level, and health condition (raw scores and controlled for age and gender by ANCOVAs (=in bold)).

	PF	RP	BP	GHP	EF	SF	RE	EWB	PHC	MHC
Marital status										
*p*-Value	0.862	0.118	0.387	0.467	0.051	0.517	0.189	0.739	0.277	0.261
Single	91.58	82.75	83.81	62.37	55.76	83.83	83.15	62.02	52.00	46.61
	**89.75**	**79.76**	**83.16**	**62.12**	**56.02**	**83.05**	**82.92**	**62.35**	**51.39**	**46.61**
Married/Cohabitant	88.84	77.05	81.33	61.20	56.08	84.22	78.54	62.80	50.67	46.66
	**89.19**	**77.62**	**81.43**	**61.23**	**56.01**	**84.36**	**78.56**	**62.73**	**50.79**	**46.65**
Widows/Widowers	83.33	93.33	82.93	66.53	63.33	88.33	93.33	66.40	52.57	49.94
	**86.65**	**98.99**	**85.23**	**67.77**	**64.17**	**90.46**	**95.24**	**66.63**	**53.95**	**50.51**
Divorced/Separated	90.46	84.09	73.73	59.18	49.09	80.68	81.82	64.00	50.38	45.61
	**91.18**	**85.27**	**73.93**	**59.24**	**48.92**	**80.96**	**81.83**	**63.83**	**50.61**	**45.58**
Education level										
*p*-Value	0.504	0.659	0.232	0.747	0.679	0.663	0.285	0.123	0.548	0.355
Elementary school education	83.22	67.22	77.58	60.02	55.22	80.56	68.89	61.78	48.46	45.29
	**86.40**	**71.34**	**78.99**	**61.19**	**55.07**	**81.74**	**69.68**	**60.78**	**49.55**	**45.22**
Middle school education	89.04	75.53	81.13	63.81	57.02	86.17	82.98	62.30	50.88	47.17
	**91.01**	**77.93**	**81.54**	**64.25**	**56.35**	**86.55**	**82.91**	**61.27**	**51.45**	**46.86**
High school education	89.08	78.18	79.78	61.26	55.21	84.21	78.69	61.52	50.64	46.22
	**89.51**	**78.79**	**80.10**	**61.50**	**55.35**	**84.46**	**78.95**	**61.50**	**50.82**	**46.29**
University education	89.88	79.66	83.20	61.52	56.52	84.27	80.75	63.73	51.31	47.02
	**89.22**	**78.80**	**82.87**	**61.25**	**56.51**	**84.00**	**80.54**	**63.69**	**51.08**	**47.02**
Health condition										
*p*-Value	0.101	0.858	**0.001 ***	**0.040 ***	**0.034 ***	0.659	0.680	0.479	**0.018 ***	0.337
No disease and Current health problem	90.62	79.52	84.06	62.89	57.13	84.38	80.63	63.17	51.67	47.04
	**90.30**	**78.99**	**83.98**	**62.91**	**57.32**	**84.31**	**80.60**	**63.31**	**51.58**	**47.09**
Past or Current disease only	86.86	76.94	80.94	60.98	55.27	85.07	77.04	63.02	50.36	46.63
	**87.14**	**77.44**	**80.71**	**60.72**	**54.67**	**84.93**	**76.70**	**62.63**	**50.38**	**46.41**
Current health problem only	88.50	76.90	77.14	58.92	54.55	82.65	79.17	61.46	49.81	45.92
	**88.69**	**77.19**	**77.42**	**59.10**	**54.81**	**82.85**	**79.50**	**61.61**	**49.92**	**46.04**
Disease and Current health problem	87.95	78.77	80.95	60.34	54.80	85.10	80.37	62.69	50.58	46.65
	**88.97**	**80.50**	**81.16**	**60.24**	**54.14**	**85.30**	**80.46**	**62.20**	**50.88**	**46.47**

PF, physical functioning; RP, role limitation, physical; BP, bodily pain; GHP, general health perception; EF, energy/fatigue; SF, social function; RE, role limitation, emotional; EWB, emotional well-being; PHC, physical health composites; MHC, mental health composites. * *p*-Value < 0.05, *p*-Value apply for adjusted scores for age and gender.

## Data Availability

Available upon reasonable request.

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
