# Peer review of "Health-Related Quality of Life According to Sociodemographic Characteristics in the South Korean Population"

_ijerph, 2022, doi:10.3390/ijerph19095223_

Round 1
Reviewer 1 Report
There is a paper that assess level of Health-related quality of life (HRQoL) in general population of South Corea. It’s well structured and results are clearly showed.
I have some points:
Material and Methods
Data Collection
- Which is the method to recruite randomly the citizens included in the study?
- The participants of the study signed a consent?
Discussion
Regardless of cultural explanations, authors propose any change in scoring for items involved in the EF and EWB dimensions, according with the low Cronbach’s alpha obtained?
I miss some comparison with data obtained in other general populations studies in other countries, including similar cultural countries. It is only stated in the conclusion.
References
There are some more recent references in the literature about HRQoL..
Reviewer 2 Report
Thank you for the opportunity to review the original article entitled Health-related quality of life using the RAND-36 Short Form health survey in the general South Korean population.
I appreciate the interesting and an important topic of this study.
Few changes could improve the quality of this paper and I have few suggestions for the Authors.
- Aim of the study: The goal should be uniform throughout the work.
Abstract
This study evaluated the HRQoL of the general South Korean population using the RAND- 2 36 questionnaire and compared HRQoL across sociodemographic characteristics.
Main. Text
(…) we aimed to obtain general population HRQoL-related data for South Korea using the RAND-36 survey and identify the factors that influence HRQoL.
- Material and Methods.
This section needs to be improved.
Inclusion/exclusion criteria could be more clearly stated and would be worth adding the information how the sample size was calculated.
Line 95
The RAND-36 is a brief self-administered questionnaire.
Are you sure? In my opinion, it is a ready standardized tool. It would be good to mention the name(s) of the author(s).
Lines 100-101
There is information about the source of the tool (https://www.rand.org/health-care/sur- 100 veys_tools/mos/36-item-short-form.html ) and the reference should be added.
- Discussion
The overall quality of the discussion sections is quite satisfactory however there is quite old literature. Would be worth adding a few (2-3) latest articles e.g. PMID: 34347192
- References
This section needs to be corrected according to the instruction for the authors.
The last position in the references is not written properly – check it, please.
OECD (2017), Health at a Glance 2017: OECD Indicators, OECD Publishing, Paris, https://doi.org/10.1787/health_glance-2017-en.
Reviewer 3 Report
Thank you for an interesting study with a public health perspective that provides new knowledge about HRQoL in a population in South Korea.
The study has several merits and contributes with new insights. However, there are some critical questions that need to be considered.
My main question is related to the study population. The study population is described as a general population. The results are suggested to be generalizable, and the authors claim that this is one of the main merits of the study. Moreover, that results from this study can be used “to evaluate the HRQoL of patients with specific diseases and compare the results with those of the general population”.
Based on the description of the study population in the manuscript it is not obvious that this population can be considered as a true population sample that is representative for the general South Korean population.
- There is a need for a more comprehensive description of how the distribution in different subgroups in the sample (i.e., study population) corresponds to the distribution in the general population. The demographic characteristics of the respondents is well described but there is no information as to whether these figures reflect the population (in 2015-2019).
There is also a need for comprehensive information about the data collection and recruitment of participants. Participants are described to be randomly recruited from a “variety of sources”.
- How were these recruitment sources defined? On what grounds were these specific sources
selected?
- How was the randomization process performed when recruiting the participants in these various sources? Was the randomization performed exactly in the same way for all sources?
- The data collection was ongoing for a very long time (four years). What was the reason for that?
- The long recruitment process and the various recruitment sources, what are the authors reflections about possible sources of bias?
Please explain how the results can be used to confirm changes in “HRQoL before and after some corona virus treatments”. From the description, no such information was collected from the participants about a corona virus infection/illness/treatment (which could have been possible only for the last months of the recruitment period).
Reviewer 4 Report
This study evaluated the health-related quality of life among adults living in South Korea. While the topic addresses an important public health issues in a high-income, rapidly aging country, the authors would add additional information and discussion so that readers can understand the context of this study and consider future study needs.
- Methods section may need to have more information according to standard guidelines for cross-sectional studies, although these guidelines may not be a mandatory requirement for the journal (https://www.equator-network.org/reporting-guidelines/strobe/ and https://www.strobe-statement.org/)
- Study design: It could be assumed that this study is conducted under the cross-sectional design. However, it could be clarified at the beginning of the Methods section.
- Eligibility of participants: Are there any other inclusion criteria for this study, in addition to “South Korean citizens aged 19–80 years?”
- Sample size: It would be questionable if the analysis had sufficient power given the sample size of this study. The authors may need to explain how the sample size was determined and justify it.
- Recruitment: The authors claimed that participants were randomly recruited from variety of sources. This point should have been described in an in-depth manner so that readers can understand the sources of biases. It the descriptions could be very long, they could be described in an appendix file.
- Data collection: How were the data collected: face-to-face interview, mail survey, online survey, or any other means? Is the questionnaire only in the Korean language?
- Data analysis: The authors employed ANCOVA to investigate factors associated with a category of health-related quality of life. Why did the authors choose ANCOVA, not a multiple regression analysis? The authors may add a brief explanation on advantages of ANCOVA. The authors involved ANCOVAs for the association between a covariate and a subcategory of health-related quality of life. Don’t the authors need to consider multiple comparisons?
- Internal consistency: As a potential reason of low internal consistencies regarding EWB and EF, the authors discussed that “items 24, 25, and 28, which are part of the EWB dimension and concern the negative aspects of mental health, the respondents may have selected responses that portrayed their mental health more positively, regardless of their actual condition.” It sounds like RAND-36 and any other quality of life scales that contain negative aspects of wellbeing may not be relevant as psychometric scales in the South Korean context. This point raises concerns about the reliability of measurement. Do the authors really think “These HRQoL data for the general population can be used in future studies that seek to use the RAND-36 questionnaire to evaluate the HRQoL of patients with specific diseases and compare the results with those of the general population?” It sounds like alternative measurement is called for to better capture health-related quality of life for the general population in South Korea. The authors may want to discuss more about the relevance of this measurement.
- Causal inference: The authors used the words that inferred causal relations between the exposure and outcome, such as “effect.” They may consider avoiding these words as the study design of this study might not allow the authors to infer such causal relations.
Round 2
Reviewer 3 Report
Thank you for the revised version of the manuscript. My main question was related to the study population and that the population was described as a general population. This has been corrected in many parts of the manuscript. However, the authors are still describing the study population as a general population in some vital paragraphs. See below. Please also consider the description in these paragraphs.
Keywords. General population
r.271. “First of all, we did not occur sampling before recruitment of the general population”.
r.296. “These HRQoL data 296 for the general population can be used in future studies……. “
